# Effect of Vacuum Heat Treatment on Larch Earlywood and Latewood Cell Wall Properties

**Bailing Sun, Yamei Zhang, Yingying Su, Xiaoqing Wang and Yubo Chai \***

Research Institute of Wood Industry, Chinese Academy of Forestry, Beijing 100091, China
\* Correspondence: chaiyubo@caf.ac.cn; Tel.: +86-010-62889462

**Abstract:** The aim of this study was to evaluate the hygroscopicity and nanomechanics of earlywood (EW) and latewood (LW) larch after thermal modification under vacuum conditions. Wood samples were heat-treated in a vacuum atmosphere at 180–220 °C for 6 h, then their cell wall properties were observed using dynamic water vapor sorption (DVS), imaging Fourier-transform infrared (FTIR) microscopy, and nanoindentation. The results showed that the vacuum heat treatment reduced the hygroscopicity of EW and LW and increased hysteresis between the adsorption and desorption branches of the isotherm. Compared with EW, the treatment temperature had a more pronounced influence on the hygroscopicity of LW. The Hailwood-Horrobin model was found to accurately fit the experimental data. Imaging FTIR microscopy revealed degradation of hemicellulose, cross-linking, condensation reactions, and redistribution of lignin in the cell wall. The elastic modulus for the heat-treated EW and LW cell walls increased at first and then decreased as the treatment temperature increased; the increase in LW was more intense than that in EW. Cell wall hardness also markedly increased after heat treatment. Our analysis suggests that vacuum heat treatment decreases hygroscopicity and alters the chemical composition distribution of cell walls, thus improving wood cell wall mechanics.

**Keywords:** vacuum heat treatment; larch; cell wall; hygroscopicity; nanomechanics





## 1. Introduction

As biopolymer composites with cellular structures, wood materials feature porosity, hygroscopicity, and anisotropy. Larch (*Larix Kaempferi* Carr.) is an important plantation tree species in northeast China. Due to its high hardness, aesthetically pleasing texture, and strong decay resistance, it is often used for construction, furniture, and decorative products [1]. However, the use of larch wood is restricted by its lack of dimensional stability. It is prone to cracking during drying and utilization, particularly the formation of round shake and radial cracks. As a coniferous wood, tracheid cells account for approximately 90% of the composition of larch, providing multiple functions for water conduction and mechanical support. Larch contains two main types of tracheids: earlywood (EW) tracheids, which have thin cell walls and large cell cavities, and latewood (LW) tracheids, which have thick cell walls and small cell cavities. There is an abrupt transition from EW to LW [2]. In a transverse section of air-dried larch wood, round shake often appears between EW and LW. Few previous studies have investigated the effect of EW and LW tracheid modification on larch wood dimensional stability.

Heat treatment is an environmentally friendly, physical modification method widely used to improve the dimensional stability and durability of wood [3–5]. Variations caused by heat treatment in the chemical components, physical properties, and mechanical properties of wood have been extensively investigated [6–8]. Changes in the properties of heat-treated wood are closely related to the wood species and heat treatment process parameters, such as the atmospheric conditions (e.g., vacuum, inert gas, steam, vegetable oil), temperature, and duration. The treatment of wood at high temperatures reduces the

macro-mechanical strength [4], which limits the use of heat-treated wood in most structural applications. The changes in mechanical properties are mainly attributed to the degradation of hemicellulose [9,10]. In our previous studies [11–13], we found that vacuum heat treatment limited the decrease in macro-mechanical properties while improving the dimensional stability of wood. However, the changes in chemical components across different morphological areas of EW and LW cell walls during vacuum heat treatment remained unclear. Likewise, the relationships between these changes and cell wall hygroscopicity and nanomechanics are not fully understood. The effect of heat treatment on wood cell wall mechanics has been studied under oxidizing or nitrogen conditions [14–16], where cell walls became harder after heat treatment.

In the present study, we investigated how vacuum heat treatment affected the hygroscopic and micro-mechanical properties of larch EW and LW cell walls. The water vapor sorption behavior was determined using dynamic vapor sorption equipment and a Hailwood-Horrobin model was applied to analyze monolayer and polylayer water in EW and LW cell walls. Imaging FTIR microscopy was used to track the chemical changes in cell walls. Cell wall nanomechanics were also observed via nanoindentation. The mechanism of vacuum heat treatment on water sorption behavior and the nanomechanics of softwood were determined to provide optimum technological parameters for treatment.

## 2. Materials and Methods

### 2.1. Materials

Larch (*Larix kaempferi* Carr.) wood was harvested in Liaoning, China, and a disk approximately 3 cm in thickness and 31 cm in diameter at breast height was obtained. The disk was divided into six samples. Four adjacent samples were selected, three of which were modified by heat treatment under different conditions. One block was taken at the 10th growth ring (approximately 4 mm wide) of each wood sample (Figure 1), which was sliced into EW and LW for DVS analysis, nanoindentation, and imaging FTIR microscopy.

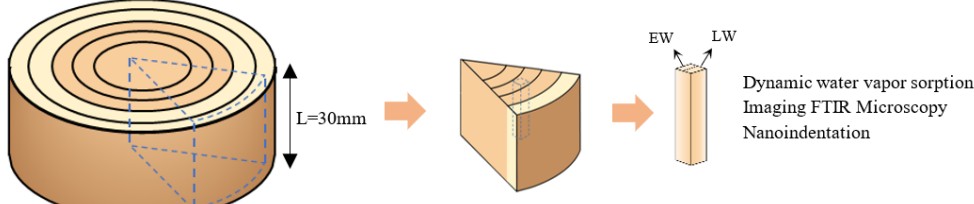

**Figure 1.** Wood sampled from disk of *Larix kaempferi* Carr.

### 2.2. Vacuum Heat Treatment

The wood samples were firstly dried at $103 \pm 2\,°C$ to constant weight, then placed into a vacuum heat treatment box (HJ-ZK60, Hengjun Instrument, Shandong Province, China) for modification. The treatment parameters were 180, 200, and 220 °C for 6 h under negative pressure ($-0.02$ to $-0.08$ MPa) [17]. The selection of processing parameters was based on the results of previous experimental research on larch where physical and mechanical properties were effectively improved [2].

### 2.3. Dynamic Water Vapor Sorption (DVS) Analysis

The wood sorption behavior was analyzed using DVS equipment (IGAsorp, Hiden Isochema Ltd., Warrington, UK) with liquid deionized water. All wood samples were cut into long sticks [5 mm × 1 mm × 1 mm (L × R × T)] and oven-dried. Approximately 20 mg of wood was placed on the sample holder in the DVS microbalance. The sorption isotherms were determined over a relative humidity (RH) range from 0 to 95% at $25 \pm 0.1\,°C$. Changes in weight of the wood sample were determined on an electronic microbalance (0.1 μg accuracy). When the mass variation in the sample (dm/dt) was less than 0.002% per minute over a 600 s period, the instrument maintained a constant target RH. The results reported here are the average values of three repetitions.

We used the Hailwood-Horrobin (H-H) model to predict the monolayer and polylayer water of wood [18,19]:

$$M = M_h + M_d = \frac{1800}{w} \left( \frac{K_1 K_2 H}{100 + K_1 K_2 H} + \frac{K_2 H}{1 - K_2 H} \right) \tag{1}$$

where $M$ is the wood moisture content (MC) at a given RH percentage ($H$). $M_h$ and $M_d$ are the MC related to monomolecular sorption (hydrate water) and polymolecular sorption (dissolved water), respectively. $W$ is the molecular weight of the cell wall polymer at the sorption site per mole of water. $K_1$ and $K_2$ are the equilibrium constants separately associated with the hydrate formed from dissolved water and dry wood, as well as dissolved water and water vapor.

### 2.4. Imaging Fourier Transform Infrared (FTIR) Microscopy

Using attenuated total reflection (ATR) mode, we obtained FTIR images on a Spectrum Spotlight 400 FTIR microscope connected to a Spectrum FTIR Frontier spectrometer (Perkin Elmer Inc., Waltham, MA, USA). The transverse surface of a sample with dimensions of 4 mm × 4 mm × 3 mm (R × T × L) were cut flat using a frozen section machine (CM3050 S; Leica, Wetzlar, Germany). Measurements provided a pixel resolution of 1.56 × 1.56 μm$^2$. The FTIR spectra were recorded at a spectral resolution of 4 cm$^{-1}$ between 4000 and 750 cm$^{-1}$. An average of 32 scans per pixel was used to increase the signal-to-noise ratio. The size of the FTIR absorbance image was 100 × 100 μm$^2$, which was calculated based on the average absorbance of the whole infrared range after atmospheric compensation. Images on transverse sections were selected at random and at least 10 areas from each sample were recorded. The selected images were processed using Spotlight 1.5.1 and Spectrum 6.3.5 software (Perkin Elmer Inc.).

### 2.5. Nanoindentation (NI)

EW and LW samples for the NI experiment were cut to approximately 20 mm × 7 mm × 7 mm (L × R × T) dimensions. A tilt apex in the LW or EW was created using a knife, then the transverse surface was smoothed with an ultramicrotome equipped with a diamond knife. The smoothed samples were adjusted in the nanoindenter room for at least 1 day. The NI experiment was performed on a Hysitron TI 950 nanomechanical device with a Berkovich-type diamond indenter tip. The maximum force applied was 200 μN at a constant load for 5 s. The load was held for 2 s, then unloaded for 5 s.

Each sample was examined under a microscope, as shown in Figure 2. Indentations were made on the middle width of the early and late tracheid cell walls, so we assumed that the S-2 layer was detected (Figure 2c,f). At least five tracheids were examined for each untreated and heat-treated wood sample, ultimately yielding 25 measurements for the wood under each modification condition [20]. The elastic modulus ($E_r$) and hardness ($H$) were calculated from the load-displacement data with reference to Oliver and Pharr [21]. $H$ was determined as follows:

$$H = \frac{P_{max}}{A} \tag{2}$$

where $P_{max}$ is the peak load. $A$ is the corresponding contact area. $E_r$ is calculated as follows:

$$E_r = \frac{dP}{dh} \frac{1}{2} \frac{\sqrt{\pi}}{\sqrt{A}} \tag{3}$$

where $dP/dh$ is the slope of the tangent of the initial unloading curve in the load-displacement diagram. The elastic modulus ($E$) of sample was determined as follows:

$$\frac{1}{E_r} = \frac{1 - v^2}{E} + \frac{1 - v_i^2}{E_i} \tag{4}$$

where $E_i$ and $v_i$ are the elastic modulus and Poisson's ratio of the tip, which are 1141 GPa and 0.07 for diamond tips, respectively. $E$ and $v$ are the elastic modulus and Poisson's ratio of the sample. We assumed that the Poisson's ratio of wood cell wall was 0.25 [22].

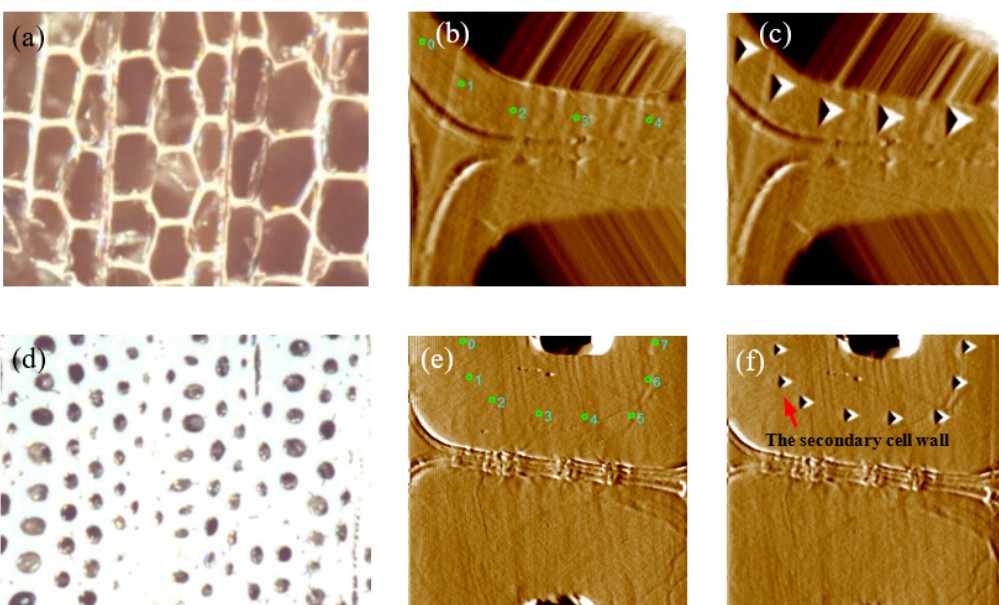

**Figure 2.** NI test area (**a**,**d**), test points and indentation after unloading (**b**,**c**,**e**,**f**). EW (**a**–**c**), LW (**d**–**f**).

## 3. Results and Discussion

### 3.1. Sorption Behaviors of Untreated and Heat-Treated Wood

As a hygroscopic material, wood is sensitive to changes in ambient humidity, including adsorbing and desorbing water as RH increases or decreases [23]. Figure 3 shows the sorption isotherms of EW and LW before and after heat treatment. All the sorption curves were similar with a characteristic sigmoidal shape. The equilibrium moisture content (EMC) of all the samples increased with increasing RH. Heat treatment reduced EMC for both the adsorption and desorption loops of the isotherm. EMC also decreased as treatment temperature increased. At 220 °C, the EW and LW curves reached their lowest points. Vacuum heat treatment appeared to reduce the wood hygroscopicity, mainly due to a reduction in hydrophilic groups. In addition, we found substantial differences in the adsorption isotherms between EW and LW. Untreated and heat-treated EW both showed higher EMC values than LW. When the EW was heat-treated at 220 °C, its hygroscopicity significantly changed and was similar to that of untreated LW.

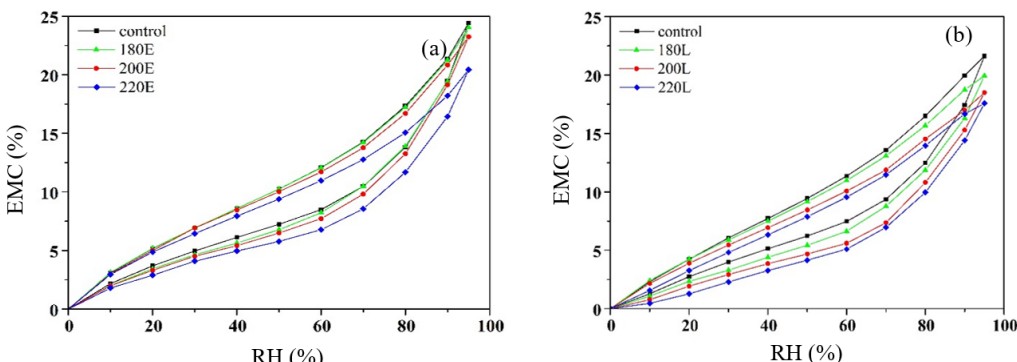

**Figure 3.** Moisture sorption isotherms of untreated and vacuum heat-treated EW (**a**) and LW (**b**).

Figure 4 shows the hysteresis of untreated and vacuum heat-treated EW and LW. Heat treatment led to an increase in hysteresis compared to the untreated wood. Similar results have been previously reported [24,25]. Hysteresis increased in EW as treatment

temperature increased, but there was no obvious effect on sorption hysteresis among heat-treated LW samples at different treatment temperatures. Thus, the sorption hysteresis may be related to the state of the specimens and the process parameters of heat treatment.

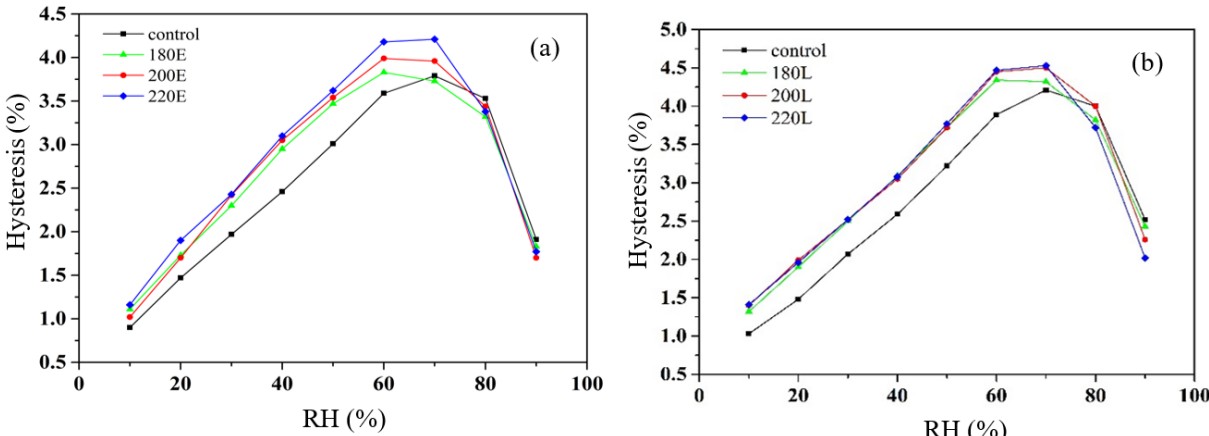

**Figure 4.** Sorption hysteresis of untreated and vacuum heat-treated EW (**a**) and LW (**b**).

The adsorption isotherm data of untreated and heat-treated EW and LW was curve-fitted using the H-H model. The fitting degree was above 0.996 (Table 1). We deconvoluted the sigmoidal isotherms into monolayer ($M_h$) and polylayer ($M_d$) water, as shown in Figures 5 and 6, where it is apparent that heat treatment reduced the $M_h$ and $M_d$ of wood. Unlike in EW, heat treatment significantly decreased the monolayer water in LW, reaching equilibrium at lower RH with increasing treatment temperature. The intersection of monolayer and polylayer water curves for LW also markedly decreased at lower RH; when RH was higher than the intersection point, polylayer water comprised the majority of the water in the sample.

We determined the values for $M_h$, $M_d$, and total water ($M_h + M_d$) at 100% RH using the fitting parameters (Table 2). $M_h$, $M_d$, and $M_h + M_d$ values decreased in heat-treated wood as treatment temperature increased. These reductions were largest after 220 °C treatment, where $M_h$, $M_d$, and $M_h + M_d$ were 18.7%, 14.7%, and 15.2% lower for EW and 49.4%, 13.5%, and 17.9% lower for LW, respectively. This implied that the vacuum heat treatment decreased the number of major sorption sites (OH groups) of cell walls to improve the wood's dimensional stability.

**Table 1.** Hailwood-Horrobin fitting parameters.

| | Sample | $K_1$ | $K_2$ | $W$ | $R^2$ |
|---|---|---|---|---|---|
| | control | 9.64 | 0.862 | 400.7 | 0.9991 |
| EW | 180 °C | 6.75 | 0.858 | 393.7 | 0.9997 |
| | 200 °C | 7.37 | 0.866 | 421.7 | 0.9987 |
| | 220 °C | 8.55 | 0.866 | 486.5 | 0.9988 |
| | control | 4.61 | 0.849 | 415.4 | 0.9991 |
| LW | 180 °C | 2.75 | 0.838 | 415.2 | 0.9994 |
| | 200 °C | 2.25 | 0.853 | 474.4 | 0.9967 |
| | 220 °C | 0.89 | 0.838 | 441.6 | 0.9984 |

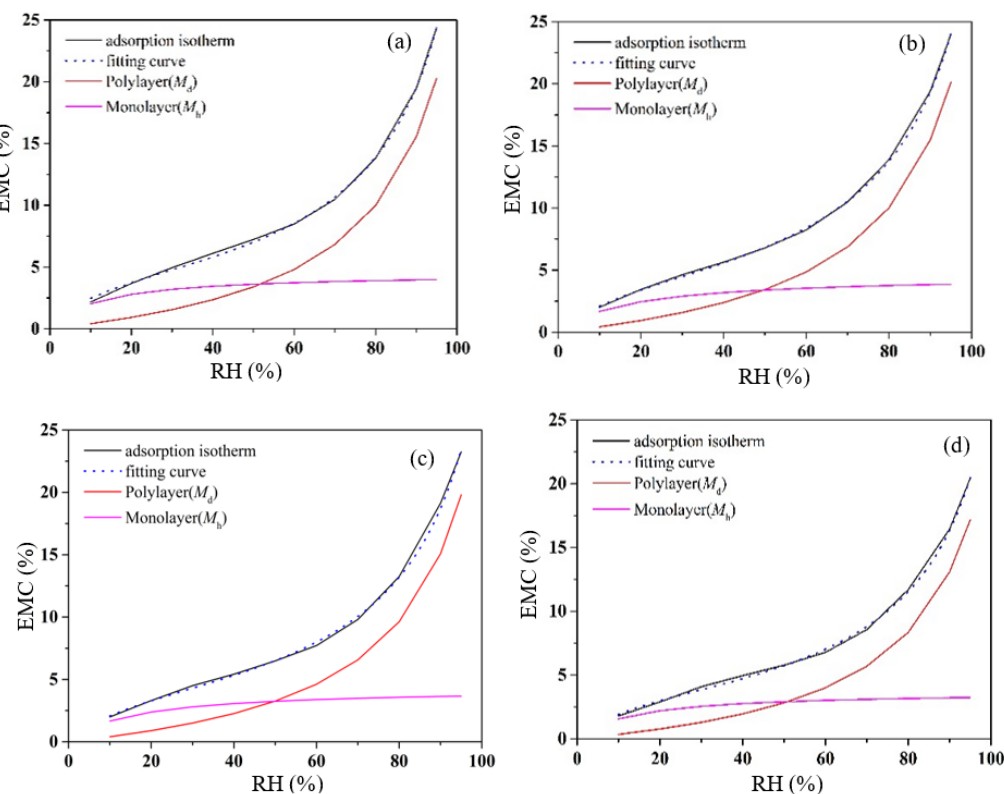

**Figure 5.** Monolayer and polylayer water determined using the H-H model, and the best fit line of sum of monolayer and polylayer adsorption isotherm using the adsorption isotherm data. Control EW (**a**), 180 E (**b**), 200 E (**c**), 220 E (**d**).

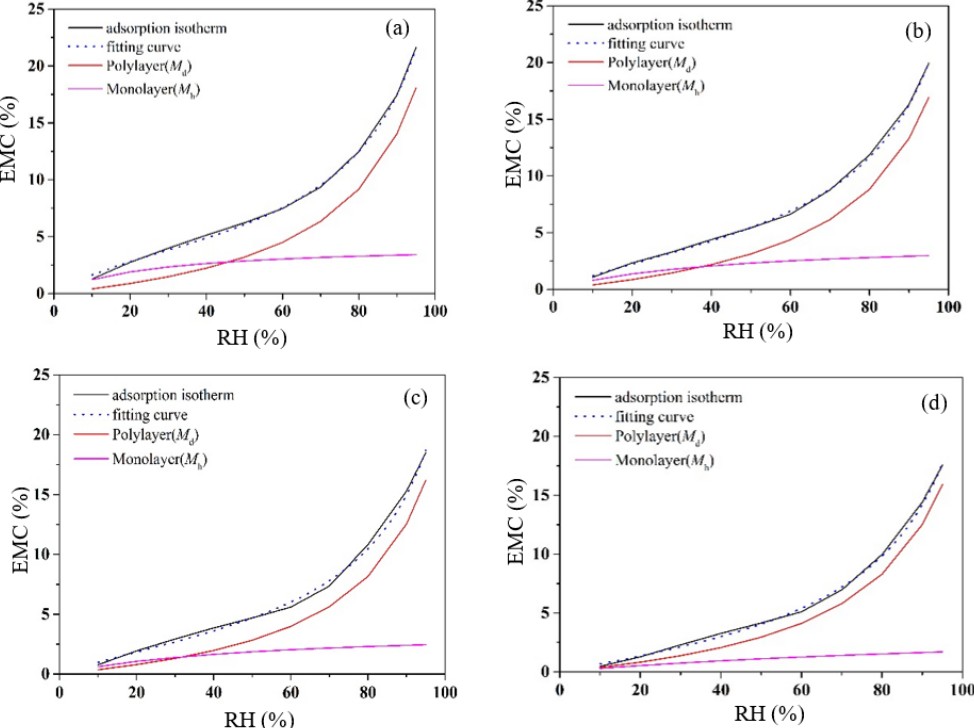

**Figure 6.** Monolayer and polylayer water determined by the H-H model and best fit line of sum of monolayer and polylayer adsorption isotherm by adsorption isotherm data. Control LW (**a**), 180 L (**b**), 200 L (**c**), 220 L (**d**).

**Table 2.** $M_h$ and $M_d$ obtained from H-H fits projected to 100% RH.

| Sample | | $M_h$ (%) | $M_d$ (%) | $M_h + M_d$ (%) |
|---|---|---|---|---|
| EW | control | 4.00 | 28.05 | 32.05 |
| | 180 °C | 3.90 | 27.62 | 31.52 |
| | 200 °C | 3.32 | 26.96 | 30.28 |
| | 220 °C | 3.25 | 23.91 | 27.17 |
| LW | control | 3.45 | 24.36 | 27.81 |
| | 180 °C | 3.02 | 22.42 | 25.44 |
| | 200 °C | 2.49 | 22.01 | 24.50 |
| | 220 °C | 1.74 | 21.08 | 22.82 |

### 3.2. Imaging FTIR Spectra of Untreated and Heat-Treated Cell Walls

The FTIR spectra of tracheid cell walls of EW and LW before and after vacuum heat treatment are shown in Figure 7a,b. The bands in the $3800-2750$ cm$^{-1}$ region were attributed to $-OH$ and $-CH_3/-CH_2-$ stretching vibrations. For heat-treated EW and LW, the intensity of the absorbance at approximately 3336 cm$^{-1}$ decreased with increasing treatment temperature. This suggested that a reduced number of OH bands decreased the hygroscopicity of heat-treated wood, which was consistent with our DVS results. The band intensity related to C=O vibration of the non-conjugated acetyl or carbonyl groups in xylan at 1734 cm$^{-1}$ decreased [26], which indicated that some acetyl groups in the heat-treated wood cell walls were deacetylated. We also found that the band intensity at 1636 cm$^{-1}$ decreased, which was attributed to C=O stretching of lignin [27]. This change may reflect a cross-linking reaction in lignin during heat treatment [28]. The band intensity at 1508 cm$^{-1}$ slightly increased, which was related to the stretching vibrations of the C=C bonds of aromatic skeletal in lignin [26], indicating that the relative amount of lignin in wood cell wall increased with temperature. The intensity of the bands near 1732 cm$^{-1}$ and 1029 cm$^{-1}$, related to $C-H$ bending vibrations and $C-O$ stretching vibration in cellulose and hemicelluloses, markedly decreased at 220 °C for EW and LW. We attributed this change to the thermal degradation of hemicellulose.

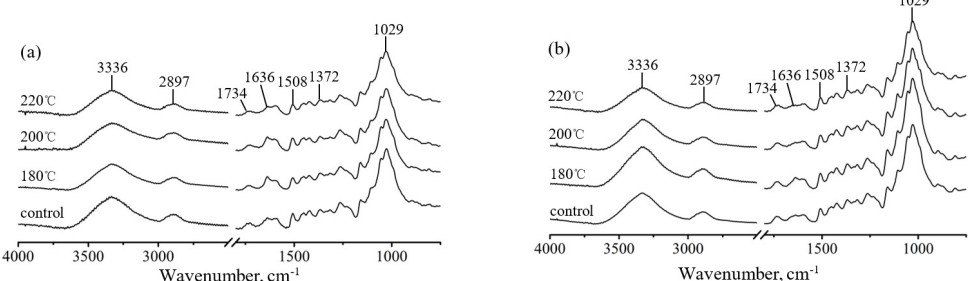

**Figure 7.** FTIR spectra of tracheid cell wall of EW (**a**) and LW (**b**) before and after vacuum heat treatment.

We also gathered images of the total average absorbance of the untreated and heat-treated EW and LW. Spectral images were generated with the bands at 1734 cm$^{-1}$ and 1508 cm$^{-1}$ related to C=O vibration in hemicellulose and aromatic skeletal vibrations in lignin, respectively (Figures 8 and 9). As shown in Figures 8C,G,K,O and 9C,G,K,O, the intensities of each pixel in the spectral images generated using the 1734 cm$^{-1}$ peak decreased as treatment temperature increased, indicating that hemicellulose degraded in the cell wall. Lignin appeared to be mainly distributed in the intercellular layer in untreated EW and LW (Figures 8D and 9D). The relative content of lignin in the cell wall increased with increasing temperature. After 220 °C treatment, the distribution area with a relatively high concentration of lignin in the cell walls increased. This change could be attributed to the degradation of carbohydrates and cross-linking, ramification, and redistribution of lignin [29–31].

Control

180℃

200℃

220℃

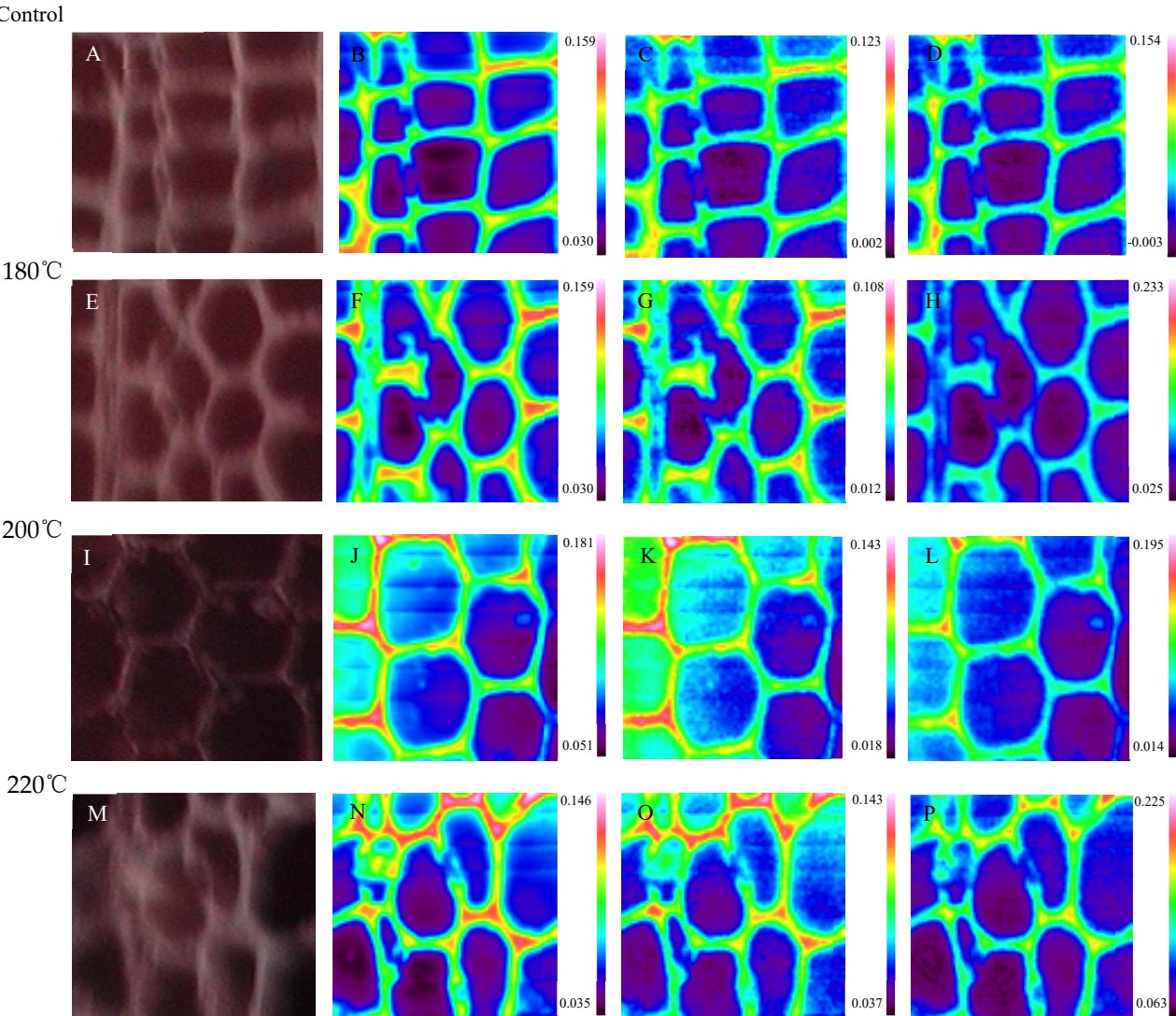

**Figure 8.** Visual images (**A,E,I,M**), total absorbance images (**B,F,J,N**), and single wavenumber images of EW before and after vacuum heat treatment. Spectral images generated with the peak at 1734 cm$^{-1}$ (**C,G,K,O**), spectral images generated with the peak at 1508 cm$^{-1}$ (**D,H,L,P**).

### 3.3. Elastic Modulus and Hardness of Untreated and Heat-Treated Cell Walls

Table 3 shows the elastic modulus ($E_r$) and hardness ($H$) of the tracheid cell walls of untreated and heat-treated EW and LW. The $E_r$ values in untreated EW and LW tracheids were 16.1 and 17.8 MPa, respectively. The $E_r$ for the heat-treated EW and LW increased at first and then decreased as treatment temperature increased. The increase in $E_r$ was significant after 180 °C treatment, at 16.8% (EW) and 20.8% (LW), and reached 4.3% (EW) and 19.1% (LW) after 220 °C treatment. Heat treatment markedly affected the $E_r$ of LW. The $H$ of EW and LW tracheids increased with treatment temperature. Compared with untreated wood, the $H$ for heat-treated EW and LW increased by 15.3%–25% and 15.4%–26.6%, respectively, at 180–220 °C for 6 h. We found no significant differences between EW and LW in this case. Previous studies have shown similar results [15,32].

The mechanical properties of wood cell walls are related to the moisture content, chemical composition, crystallinity, and other properties of the material [33]. Previous studies have shown that filling the spaces between cellulose-hemicellulose strands with lignin increases the hardness of wood cell walls [34]. We found that vacuum heat treatment decreased the EMC and altered the chemical composition distribution of cell walls, which improved wood cell wall mechanics.

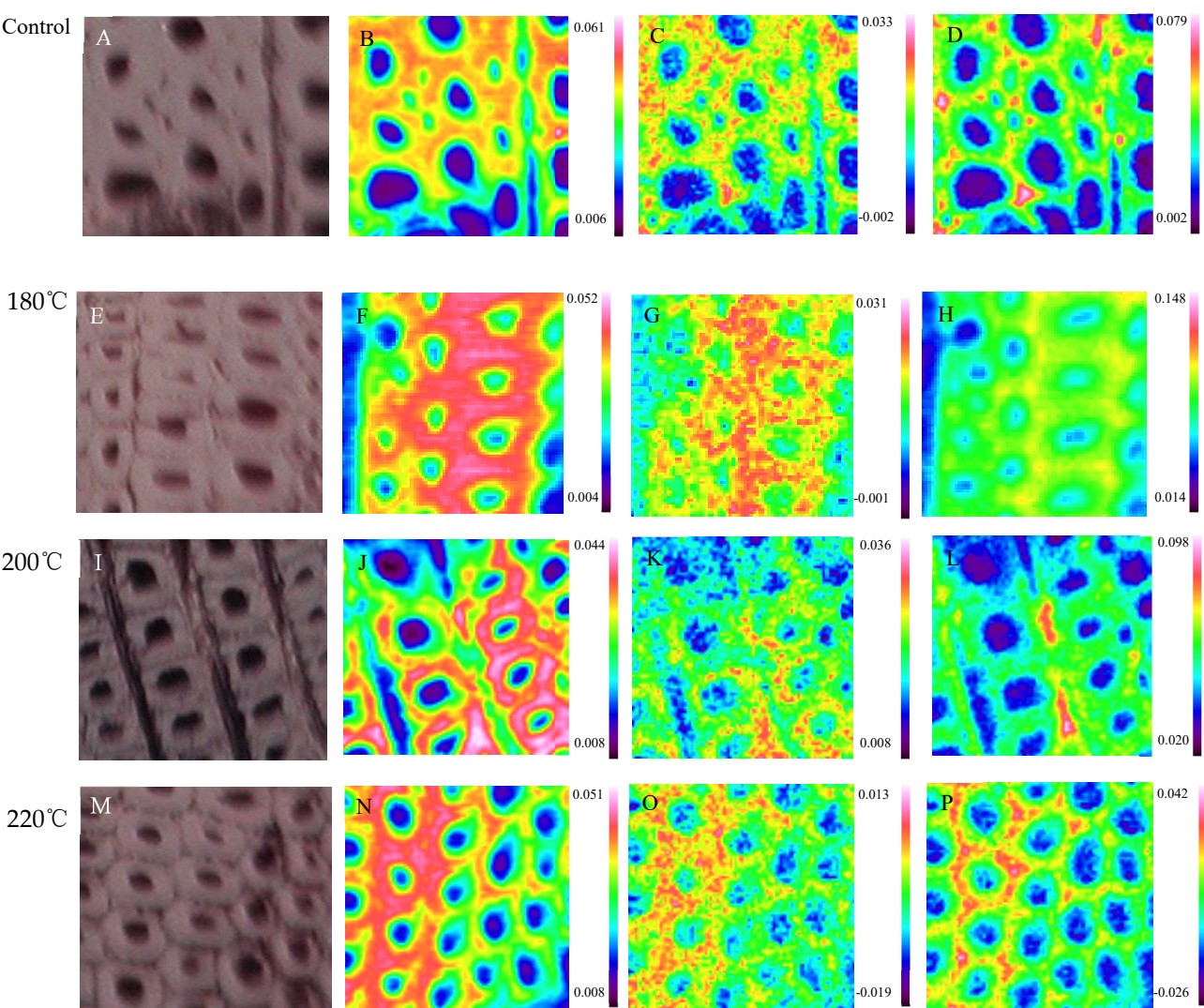

**Figure 9.** Visual images (**A,E,I,M**), total absorbance images (**B,F,J,N**), and single wavenumber images of LW before and after vacuum heat treatment. Spectral images generated with the peak at 1734 cm$^{-1}$ (**C,G,K,O**), spectral images generated with the peak at 1508 cm$^{-1}$ (**D,H,L,P**).

**Table 3.** Elastic modulus and hardness values of untreated and vacuum heat-treated wood cell wall as determined by quasi-static indentation.

| Sample | Elastic Modulus (GPa) | | Hardness (GPa) | |
|---|---|---|---|---|
| | EW | LW | EW | LW |
| control | 16.1 (1.65) | 17.8 (1.99) | 0.496 (0.061) | 0.514 (0.063) |
| 180 °C | 18.8 (1.98) | 21.5 (1.41) | 0.572 (0.052) | 0.593 (0.061) |
| 200 °C | 17.8 (1.52) | 21.3 (1.82) | 0.605 (0.048) | 0.624 (0.058) |
| 220 °C | 16.8 (1.63) | 21.2 (1.34) | 0.620 (0.053) | 0.651 (0.049) |

SD in parenthesis.

## 4. Conclusions

The vacuum heat treatment conducted in this study reduced the hygroscopicity of larch EW and LW. Compared with EW, the treatment temperature had a more pronounced influence on the hygroscopicity of LW. Heat treatment also increased hysteresis between the adsorption and desorption branches of the isotherm. The adsorption isotherms analyzed using the H-H model were in close agreement with our experimental data. The monolayer and polylayer water of heat-treated EW and LW decreased as treatment temperature increased. Imaging FTIR microscopy indicated degradation of the acetyl or carbonyl

groups in xylan and a loss of the C=O group linked to the aromatic skeleton in lignin. The spectral images showed a decrease in hemicellulose in cell walls and an increase in lignin in cell walls, together suggesting cross-linking and condensation reactions, as well as redistribution of lignin in the cell wall during vacuum heat treatment. These chemical changes improved the heat-treated wood cell wall mechanics. The cell walls of heat-treated EW and LW also hardened substantially after heat treatment; the elastic moduli varied at different treatment temperatures. The results of this work demonstrate that vacuum heat treatment can decrease the number of moisture sorption sites in wood cell walls to improve the dimensional stability and cell wall mechanics of the material. Considering the effects of vacuum heat treatment on the properties of EW and LW, optimal process parameters may markedly improve the dimensional stability of larch wood while avoiding the decrease in wood mechanical properties.

**Author Contributions:** All authors contributed to this manuscript: B.S. performed the experiments, analyzed the data, and wrote the first draft; Y.Z. and Y.S. also analyzed the data and results; X.W. and Y.C. reviewed and edited the manuscript. All authors have read and agreed to the published version of the manuscript.

**Funding:** This research was funded by the National Natural Science Foundation of China (Grant No. 31901244).

**Data Availability Statement:** The datasets generated during and/or analyzed in this study are available from the corresponding author upon reasonable request.

**Conflicts of Interest:** The authors have no relevant financial or non-financial conflicts of interest to disclose.

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
