# Peer review of "Effect of Vacuum Heat Treatment on Larch Earlywood and Latewood Cell Wall Properties"

_forests, doi:10.3390/f14010043_

Round 1

Reviewer 1 Report

First of all he  objectives of manuscript is clearly stated.The manuscript have knowledge to add to  scientific literature because is essential to invastigated the influence of Vacuum Heat Treatment on  Earlywood and Latewood Cell Wall properties.

I have to note that the section of material and methods isnt adequately described. To be more specific there isnt information about specific gravity , dimensions of samples (wood blocks)adial, tangential, longitudinal. To minimize the effects of sample variation, how groups of samples with equal weight distributions were assigned? Width of annual ring? The weight of the samples was recorded before (w0) and after each treatment (w1) to assess the weight gain, and the weight loss, due to vacuum-heat treatment? The volumes of the samples recorded before (V0) and after treatments (V1)?.

Colour measurement?

There is lack of statistical analysis in order to determine whether measured properties were significantly different among treated samples categories. Dimensional changes after treatment?

 Colour changes after treatment?

 The flexibility increased of wood cell wall at different RH changes during adsorption and desorption runs, and reduction of sorption hysteresis?

·          The paper is quite well organized.

·          The topic is interesting   

·         The  author's representation of the most relevant recent advances in the field is efficient. The references are relevant to the topic and cover both historical literature and more recent advances .

·      As for sampling I thing the quantitie isn’t enough.For future research is important to investigate t hydro-thermal modification on different species and include more parameters different durations of time for instanse instead of 6 h less time like 2-4 hours and add a gas like NH4.

Reviewer 2 Report

 The study is interesting and has scientific merit, however, one of the main problems it’s lacking of the comparison between latewood and earlywood. The justification on why the authors conducts the treatment on EW and LW is also lacking, please revise. Another problem is the whole manuscript is full of active voice, such as we determined, we found etc., which should be avoided.

Introduction is generally well-written, but more literatures could be added.

Is abrupt transition from EW to LW in Larch has something to do with its dimensional stability? Please mention it clearly in the first paragraph. As the EW and LW is the focus in this study, the problems regarding these two parts of wood should be stressed.

Line 65 – the sentence should be written in passive voice – ex: larch wood was harvested in…..

Line 73-74 – change the sentence to passive voice,

many sentences are in active voice, please change it

Line 81 – “About 20 mg of wood was put on the sample holder in the DVS microbalance” is one wood stick [5 mm× 1 mm× 1 mm] equal to 20 mg?

Figure 3 – the legend in the figure 180E, 180L is inappropriate.

The title of Figure 8 & 9 is not clear. Is it the samples treated with different temperature? Please indicate clearly.

Line 208-218 - The discussion is also vague, what is the indication for decreased hemicellulose and increased lignin? Colour? Colour intensity? Please mention clearly.

Line 234 – “The increase in Er was significant after 180 °C treatment, at 16.8% (EW) and 20.8% (LW) and reached 4.3% (EW) and 19.1% (LW) after 220 °C treatment” Increase? Please check

Reviewer 3 Report

The topic of the research work and manuscript is really interesting and provides new information. However there are several issues to be addressed towards its quality improvement before thinking of publication.

The key words are not simple enough to be potentially searched by the readers (try to avoid whole phrases). In lines 25-28, please provide a reference. In lines 33-35, is that only for the specific species? In lines 44-46, what is the reason for this phenomenon of strength reduction? please explain in the text providing more information. The state-of-the-art description is generally quite short in introduction section and the total number of references used in this manuscript is really low, given the rich literature. Try to describe the most significant findings of literature so far concerning the thermal treatment of wood and the potential improvement of wood exposed. Please, use as well the manuscriphttps://doi.org/10.5552/drvind.2021.2026 , which could help you descibe in more details the chemical changes in wood mass induced by its thermal treatment. Indeed, some phenomena are not fully understood, though you could present what is the current knowledge and how the specific work attempts to cover a knowledge gap. You would rather not use "we", it would be more preferable to use passive voice instead (last paragraph of introduction). 

In materials and methods chapter, did you follow a specific sampling method for the trunk choice? Why did you choose to treat thermally the wood samples in such pieces of so small dimensions?Is this choice close to reality of thermal treatment of wood? How could the results of the specific work be useful to be applied in other dimensions pieces of such a huge difference? Please, provide an image of the vacuum chamber and the specimens of this experimental work. Explain in the text te aim of Dynamic Water Vapor Sorption (DVS) Analysis to be more useful to the reader, as well as all the other methods to be clear and repeatable. You did not describe any statistical analysis of the results in materials-methods chapter, if any. Would that be possible to discuss further the findings of yours compared to previous findings of literature? Table 3 could be more explained/interpreted. The word hemicellulose should be used in plural since it is not only one thing. Could you make some improvement in highlighting the significance and practical meaning of application of the specific research and its findings (probably highlighted in conclusions section, as well as abstract and combined with the last paragraph of introduction).

Reviewer 4 Report

Comments and Suggestions for Authors

Dear Authors and Editors,

This paper focuses on the hygroscopicity and nanomechanics of larch earlywood and latewood after thermal modification in a vacuum. It provides results on cell wall properties observed using dynamic water vapour sorption, imaging microscopy, and nanoindentation.

The research results are original and significant, and the results justify and support the conclusions, although the article lacks the proper statistical analysis of the results.

For future research, it would be advisable to cut more than one tree (at least three) to rule out the anisotropy of the tree itself and the possibility that this tree is somehow more unusual than others.

The English language is understandable but needs some corrections.

The paper is acceptable for publishing in the journal Forests after minor corrections and some observations in the text below.

Line 65. Rewrite sentence: For this investigation, a 32-year-old larch (Larix kaempferi Carr.) wood was harvested in Liaoning, China, and a disk of approximately 3 cm thickness was sawn at breast height.

Line 67. Which tree samples were treated, and what did you do with the other three? Was it a random pick, or did you have some orienteers, like north orientation, south…?

Line 68. Why did you take the sample from the 10th growth ring?

Line 73. Rewrite sentence: First, the wood samples were dried… Then they were modified in a vacuum chamber …

Line 74. You have to specify which vacuum chamber it was. Producer, number…

Line 81. Oven-dried at which temperature and to what purpose?

Line 84. Electronic microbalance specifications?

Line 137. How many samples did you test per treatment?

Line 254. Lose “we” in sentence.

Round 2

Reviewer 1 Report

This revised version made significant improvement. I agree with the publication of the paper.

Reviewer 2 Report

The paper has been improved accordingly. The content is fine, only minor grammar checking and formatting is required.

Reviewer 3 Report

As I have checked the authors have implemented the proposed changes in the revised verion of manuscript towards the improvement of their work. Almost all the changes have been implemented and in my opinion, the manuscript is well-prepared and organized enough to be accepted for publication in this journal. I remain at your disposal for any clarification.